# Syntactic Information Extraction in the Parafovea: Evidence from Two-Character Phrases in Chinese

**DOI:** 10.3390/bs15070935

**Published:** 2025-07-10

**Authors:** Zijia Lu

**Affiliations:** School of Law, Tianjin University of Commerce, No. 409, Guangrongdao Road, Tianjin 300134, China; luzijia_psy@163.com

**Keywords:** parafoveal preview, syntactic processing, Chinese reading, eye movements

## Abstract

This study investigates syntactic parafoveal processing in Chinese reading using a boundary paradigm with two-character verb–object phrases. Participants (N = 120 undergraduates) viewed sentences with manipulated previews (identity, syntactically consistent, and inconsistent previews). Results showed a selective syntactic preview effect: syntactical violations reduced target word skipping rates, but fixation durations remained unaffected. This dissociation contrasts with robust syntactic preview benefits observed in alphabetic languages, highlighting how Chinese’s lack of morphological markers constrains parafoveal processing. The findings challenge parallel processing models while supporting language-specific modulation of universal cognitive mechanisms. Our results advance understanding of hierarchical information extraction in reading, with implications for developing cross-linguistic reading models.

## 1. Introduction

During sentence reading, readers can process information not only from the currently fixated word (n) but also partially from the adjacent rightward word (n + 1) through parafoveal preview, which plays a crucial role in efficient reading. Extensive research has investigated how lexical properties of words n and n + 1 influence eye movement behavior, word recognition, and sentence comprehension. However, the depth of parafoveal processing, particularly the extraction of high-level linguistic information, remains a contentious issue in current reading research. Existing studies have consistently demonstrated stable parafoveal preprocessing of low-level visual information, including orthographic features ([6]) and lexical/sublexical phonological information ([1]). In contrast, researchers have yet to reach a consensus regarding the extraction of high-level linguistic information (e.g., semantic and syntactic information) in the parafovea. Intriguingly, this preview effect exhibits remarkable language specificity: For semantic information, alphabetic readers (e.g., English) show difficulty in extracting parafoveal semantic information ([13], [14]), whereas Chinese readers demonstrate robust semantic preview benefits ([23], [24]; [27]). Conversely, for syntactic information, this pattern is reversed: alphabetic readers can effectively utilize parafoveal syntactic cues ([2]; [18]; [20]), while Chinese readers show limited syntactic preview benefits ([8], [9]). The cross-linguistic comparison of parafoveal preview effects are summarized in Table 1. This cross-linguistic asymmetry in high-level linguistic information processing not only holds significant theoretical implications for understanding human language processing mechanisms but also provides a unique perspective for investigating how language characteristics influence reading processes. Notably, compared to semantic preview studies, research on syntactic preview processing in Chinese remains scarce, with existing findings showing inconsistencies. Therefore, this study focuses on syntactic information preview in Chinese to systematically examine the mechanisms underlying high-level linguistic information processing in the parafovea, aiming to provide new empirical evidence for understanding language information processing during reading.

Parafoveal preview studies primarily employ the gaze-contingent boundary paradigm proposed by [10] ([10]). This paradigm involves setting an invisible boundary to the left of the target word. When readers’ saccades cross this boundary, the preview stimulus is replaced in real-time by the target word. This method allows researchers to examine whether specific information can be effectively extracted from the parafoveal region.

The field of eye movement control research currently features two major theoretical models: the Sequential Attention Shift (SAS) model and the Parallel Graded Processing (PG) model, which offer divergent explanations for parafoveal information processing mechanisms. The SAS model (exemplified by the E-Z Reader model) posits a strictly serial processing mechanism, maintaining that attentional resources can only focus on one lexical unit at a time. According to this framework, higher-level information processing (such as syntactic analysis) cannot commence until lexical access of the currently fixated word is completed, thereby rendering effective syntactic preview processing in the parafoveal region unlikely. In contrast, the PG model (including the SWIFT) adopts a parallel processing architecture, proposing that multiple lexical units within the perceptual span can receive graded attentional allocation simultaneously. This model suggests that lower processing difficulty of the currently fixated word (e.g., high-frequency or highly predictable words) leads to higher activation levels of parafoveal words, consequently enabling the processing of higher-level information (including syntactic features like word category) in the parafoveal region. The absence of morphological markers in Chinese may align more closely with the serial processing hypothesis, though this contention still requires further empirical validation.

Recent studies using the boundary paradigm have systematically investigated parafoveal processing of syntactic information in alphabetic languages, primarily examining two key dimensions: word category consistency and tense agreement. Regarding word category consistency, [2] ([2]) created syntactic violation conditions by presenting noun previews (e.g., “surgeon”) in verb positions (e.g., “confess”), finding significantly lower target word skipping rates under violation conditions. [20] ([20]) further demonstrated that category violations prolong both gaze duration and regression path duration. [18]’s ([18]) Dutch study revealed that word category consistency affects both skipping rates and fixation durations. In terms of tense agreement, [20] ([20]) manipulated tense morphology (e.g., “refuel/landed”), showing that tense-consistent preview conditions significantly increased skipping rates while reducing all fixation duration measures except total fixation time. These consistent findings demonstrate that alphabetic readers can reliably extract both categorical and temporal syntactic information parafoveally—an advantage likely attributable to their rich morphological marking system, which provides reliable syntactic cues for parafoveal processing.

Investigating syntactic preview effects in Chinese presents unique linguistic challenges. Unlike alphabetic writing systems such as English, Chinese lacks overt inflectional morphology (e.g., tense markers) and derivational morphology (e.g., affixation), constraining researchers to examine syntactic preview primarily through word category consistency. Our research team has systematically addressed this issue through two key studies ([8], [9]). The first study compared preview effects for single-character words under category-consistent (e.g., noun “城” [city]-noun “饼” [cake]) and inconsistent (noun “城”-verb “恨” [hate]) conditions, revealing no significant differences. The second study innovatively employed a word repetition paradigm to potentially enhance category information extraction through lexical repetition priming yet similarly failed to demonstrate significant syntactic preview benefits. While [26]’s ([26], [25]) findings suggest possible subtle syntactic preview influences, direct empirical evidence for robust syntactic preview effects in Chinese reading remains elusive. This cross-linguistic discrepancy likely stems from Chinese’s distinctive grammatical features—particularly its relatively fluid word category boundaries and absence of overt syntactic markers—which may fundamentally alter parafoveal processing mechanisms compared to alphabetic languages.

Current research suggests that syntactic preview effects do not follow a simple “all-or-none” pattern but rather demonstrate processing depths that may be modulated by various factors. This perspective finds support in semantic preview studies of alphabetic languages: while early research indicated limited parafoveal semantic extraction by English readers ([13], [14]), subsequent studies employing enhanced semantic relatedness ([17]) or modified orthographic features ([12]) successfully observed semantic preview benefits. These findings suggest that Chinese syntactic preview effects may similarly be subject to such modulatory mechanisms. Consequently, systematically investigating Chinese syntactic preview effects across different experimental conditions would contribute to a more comprehensive understanding of their underlying mechanisms and enable more precise estimation of their effect size variability.

Chinese reading research has demonstrated the significant impact of lexical unit length on linguistic information processing. Compared to single-character words, two-character words and phrases exhibit processing advantages due to their higher semantic transparency and lower ambiguity ([5]; [21]). These advantages manifest in two key aspects: (1) the morphological composition of two-character words and phrases enables direct semantic derivation while reducing syntactic information extraction difficulty, and (2) they decrease contextual dependency, unlike single-character words like “行” (xíng), which require context to determine their grammatical category (noun or verb).

Building on this theoretical foundation, this study implements two crucial methodological innovations: first, we pioneer the use of two-character words/verb–object phrases as experimental stimuli to overcome the polysemy issues inherent in single-character words; second, we enhance syntactic predictability by increasing target word predictability. This dual-optimization design is expected to improve the extractability of categorical information in parafoveal vision. Therefore, we hypothesize that Chinese readers can extract partial syntactic information through parafoveal preview. Our innovative approach will provide novel empirical evidence for understanding the mechanisms underlying syntactic preview processing in Chinese.

## 2. Materials and Methods

### 2.1. Participants

A total of 120 undergraduate students (21 males, 99 females; mean age = 20.6 years, *SD* = 1.56) from Tianjin Normal University participated in the experiment. All participants had normal or corrected-to-normal vision, were native Chinese speakers, and reported no significant reading impairments. Each participant received ¥10 as compensation upon completion of the experiment.

### 2.2. Design

This study employed a single-factor (preview type), three-level within-subjects design, with the conditions being (a) identity preview (baseline, where the preview matched the target word), (b) syntactically consistent preview (the preview belonged to the same grammatical category as the target but was lexically different), and (c) syntactically inconsistent preview (the preview violated the target’s grammatical category). Each participant completed trials under all three conditions, with presentation order counterbalanced to control for sequence effects.

### 2.3. Materials

The target stimuli consisted of commonly used verb-noun two-character phrases (e.g., 爬山 “climb mountains”, 弹琴 “play the piano”). The selection process was as follows: First, 42 monosyllabic nouns with a frequency higher than 2.3 occurrences per million (*M* frequency = 133.99 per million, *SD* = 159.87; *M* strokes = 8.48, *SD* = 3.52) were selected from [19] ([19], 6th ed.) and a Chinese word frequency corpus based on movie subtitles ([3]). These nouns served as the second character in the verb-noun phrases. Each noun was then paired with a high-frequency monosyllabic verb that frequently co-occurred with it (e.g., 琴 “piano” paired with 弹 “play”), forming a natural verb-noun phrase as the target word. For the preview manipulation, we kept the verb constant and modified the noun to create the three experimental conditions:

**Identity preview**: The preview matched the target word (e.g., 弹琴 “play the piano”).

**Syntactically consistent preview**: The preview was a verb-noun phrase with a semantically implausible but syntactically valid noun (e.g., 弹窝 “play nest”).

**Syntactically inconsistent preview**: The preview was a verb-verb phrase, violating both syntax and semantics (e.g., 弹答 “play answer”).

Syntactically consistent previews were designed to maintain grammatical category consistency while introducing semantic implausibility. This design allows us to isolate the effects of syntactic processing from semantic processing. Although semantic implausibility might introduce some processing difficulty, the primary focus of this study is on the syntactic level. Under the condition of maintaining semantically anomalous content, by comparing conditions with and without syntactic violations, we can more clearly identify the role of syntactic information in parafoveal processing.

To ensure comparability, the alternative nouns (*M* frequency = 134.19 per million, *SD* = 157.34; *M* strokes = 8.45, *SD* = 3.51) and verbs (*M* frequency = 135.72 per million, *SD* = 155.92; *M* strokes = 8.45, *SD* = 3.49) were matched to the target nouns in frequency, stroke count, and visual complexity. An ANOVA confirmed no significant differences in frequency (*F* (2, 125) = 0.512, *p* = 0.601) or stroke count (*F* (2, 125) = 0.138, *p* = 0.871) across conditions. Each of the 42 target nouns was embedded in a sentence frame (mean length: 18 characters), positioned centrally to avoid sentence-edge effects. The same frames housed the three preview types (see Table 2 for examples). Critically, comparing Conditions 2 and 3 allowed us to isolate syntactic processing in parafoveal preview.

Three independent participant groups (n = 15 each) performed the naturalness, predictability, and grammatical prediction ratings separately, with no overlap between norming tasks or subsequent experimental participation. The norming results revealed that:

**Sentence Naturalness Rating**: Prior to the main experiment, we conducted a sentence naturalness evaluation. Fifteen undergraduate students rated the naturalness of sentences containing identity-preview target words using a 5-point Likert scale (1 = “highly unnatural” to 5 = “highly natural”). All sentences demonstrated high naturalness (*M* = 4.36, *SD* = 0.43), with mean ratings significantly exceeding the predetermined acceptability threshold of 3.5.

**Cloze Probability Test**: A separate group of 15 participants completed a cloze probability task to assess target word predictability. When presented with sentence fragments preceding target words, participants produced the expected continuation. Results indicated moderate predictability (*M* = 52%, *SD* = 36%), confirming that target words were neither overly predictable nor completely unexpected.

**Grammatical Category Prediction**: An additional 15 participants performed a grammatical category prediction task. After reading sentence contexts containing the target verb, they predicted the lexical category of the upcoming word. Results showed that 96% (*SD* = 5.73%) of responses correctly anticipated a noun, validating our syntactic manipulation.

### 2.4. Apparatus

The experiment employed an SR Research EyeLink 1000 Plus eye-tracker (SR Research Ltd., Ottawa, ON, Canada) to record participants’ eye movements at a sampling rate of 1000 Hz. Stimuli were presented on a display monitor with a 120 Hz refresh rate and 1024 × 768 pixel resolution. Only right-eye movements were recorded during the experiment. Viewing distance was maintained at approximately 65 cm. All sentences were displayed in 25-point SimSun font, with each Chinese character occupying approximately 33 × 33 pixels (visual angle ≈ 1.1°).

### 2.5. Procedure

The experiment was conducted in a quiet laboratory setting. Participants were tested individually under the experimenter’s supervision. After adjusting the chinrest and forehead support for proper head positioning, participants received instructions to read naturally while minimizing head movements. The task instructions, displayed on-screen, read: “You will see Chinese sentences one at a time. Read each sentence carefully and press the spacebar when you finish. Some sentences will be followed by yes/no questions—press ‘F’ for ‘yes’ and ‘J’ for ‘no’”. A 3-point eye-tracking calibration was then performed (validation threshold <0.25, mean = 0.19). Following successful calibration, participants completed practice trials before proceeding to the main experiment, during which the eye-tracker recorded all eye movements and keyboard responses.

The experimental design employed a Latin square counterbalancing across three blocks, with each participant completing one block containing 14 sentences per condition (42 total experimental sentences). The session included 3 practice trials, 18 filler sentences, and 20 randomly presented comprehension questions, lasting approximately 15 min in total.

Authors used DeepSeek Chat (version DeepSeek-V3) for the purposes of translating and language polishing of the manuscript.

## 3. Results

Participants demonstrated high comprehension accuracy (*M* = 95%, minimum = 85%), confirming their engagement with the task. Following established procedures ([11]), we excluded fixations shorter than 80ms or longer than 1200ms, as these extremes may not reflect genuine cognitive processing. Additional exclusion criteria included (1) trials with fewer than three fixations per sentence, (2) data points beyond 3 standard deviations from the mean, (3) blinks during first-pass reading of the boundary or target word, and (4) premature or delayed preview changes. This resulted in the exclusion of 15.2% of total data.

Three regions of interest (ROIs) were analyzed: the pre-target character (pre-target), the two-character target word (target), and the post-target character (post-target). For the regions of interest (ROIs), we analyzed both fixation duration measures and viewing probability using the following standard eye-movement metrics ([22]): (1) first fixation duration—the duration of the initial fixation within an ROI during first-pass reading; (2) single fixation duration—the fixation duration when only one fixation occurred in an ROI during initial left-to-right reading; (3) gaze duration—the sum of all fixation durations within an ROI from first entry until the first exit; and (4) skipping rate—the probability that an ROI was completely skipped during first-pass reading. These measures were selected to capture different stages of lexical processing, from early visual encoding to later comprehension processes.

Data were analyzed using Linear Mixed Models (LMMs) in R (4.4.1; [15]) with the lme4 package (1.1-35). This approach effectively handles unbalanced designs by weighting each participant’s data per item. We specified crossed random effects for both participants and items. Fixation durations were log-transformed to ensure linearity. When maximal models failed to converge, we systematically simplified them by first removing item correlations, then item slopes, followed by participant correlations and slopes if necessary. The preview condition served as the fixed factor in all analyses.

### 3.1. The Analysis of the Pre-Target Character

The means and standard deviations of eye-movement measures for the pre-target character are presented in Table 3, with statistical test results shown in Table 4. The comparison of mean values for each measure on the pre-target character is shown in Figure 1.

No significant differences were observed across conditions for either fixation durations or viewing probability measures on the pre-target character (all |*t*|s < 0.62, *p*s > 0.54; |*z*| < 0.75, *p* > 0.45). This null effect likely occurs because: (1) the target stimulus was a two-character compound word, and (2) the critical manipulation occurred on the second character of the target word. The intervening first character of the target word may have created sufficient spatial separation to prevent parafoveal preview effects from influencing processing of the more distant pre-target character.

### 3.2. The Analysis of the Target Word

The means and standard deviations of eye-movement measures for the target word are presented in Table 5, with statistical test results shown in Table 6. The comparison of mean values for each measure on the target word is shown in Figure 2.

Identity Preview Effect: Robust identity preview effects were observed across all temporal eye-movement measures. When the preview stimulus mismatched the target word (Condition 2), readers exhibited significantly longer fixation durations compared to the identity preview condition (Condition 1): first fixation duration (*b* = 0.1, *SE* = 0.01, *t* = 7.46, *p* < 0.001, 95% CI [0.08, 0.13]), single fixation duration (*b* = 0.11, *SE* = 0.02, *t* = 6.69, *p* < 0.001, 95% CI [0.07, 0.13]), and gaze duration (*b* = 0.18, *SE* = 0.02, *t* = 10.57, *p* < 0.001, 95% CI [0.15, 0.22]). However, no significant difference emerged in skipping rates (*b* = −0.09, *SE* = 0.09, *z* = 10.57, *p* = 0.35, 95% CI [−0.27, 0.10]), suggesting that preview validity primarily influences late lexical processing stages rather than early targeting decisions.

Syntactic Preview Effect: The syntactic violation condition (Condition 3) significantly reduced skipping probabilities relative to the syntactically consistent preview (Condition 2) (*b* = −0.23, *SE* = 0.1, *z* = −2.35, *p* = 0.019, 95% CI −0.42, −0.04]), indicating readers’ sensitivity to grammatical category information during parafoveal processing. Notably, this syntactic effect was selective to viewing probability measures, as fixation durations showed no significant differences across conditions (all |*t*|s < 1.07, *p*s > 0.29). This dissociation suggests that while syntactic violations disrupt initial word targeting, they may not necessarily prolong immediate lexical access when fixations do occur.

### 3.3. The Analysis of the Post-Target Character

The means and standard deviations of eye-movement measures for the post-target character are presented in Table 7, with statistical test results shown in Table 8. The comparison of mean values for each measure on the post-target character is shown in Figure 3.

Significant spillover effects of identity preview were observed across all eye-tracking measures on the post-target character (n + 1). Compared to the identity preview condition (Condition 1), the mismatched preview condition (Condition 2) elicited significantly longer fixation durations on the subsequent character, as evidenced by first fixation duration (*b* = 0.11, *SE* = 0.02, *t* = 4.89, *p* < 0.001, 95% CI [0.06, 0.15]), single fixation duration (*b* = 0.11, *SE* = 0.02, *t* = 4.97, *p* < 0.001, 95% CI [0.06, 0.15]), and gaze duration (*b* = 0.12, *SE* = 0.02, *t* = 5.35, *p* < 0.001, 95% CI [0.07, 0.16]). Additionally, a significant reduction in skipping rate was observed (*b* = −0.18, *SE* = 0.08, *z* = −2.22, *p* = 0.027, 95% CI −0.34, −0.02]), demonstrating that preview validity influences both temporal and spatial aspects of post-target processing.

No significant syntactic spillover effects were observed on the post-target character (n+1). Comparisons between the syntactically consistent preview (Condition 2) and syntactic violation conditions (Condition 3) revealed no statistically reliable differences in either fixation durations (all |*t*|s < 0.97, *p*s > 0.33) or skipping probability (|*z*| < 0.63, *p* > 0.53), indicating that grammatical category violations in parafoveal preview did not extend their influence to subsequent word processing.

## 4. Discussion

This study systematically investigated the processing mechanisms of syntactic information in the parafoveal region during Chinese reading using the boundary paradigm. For the first time, we employed two-character verb–object phrases as experimental materials, with strict control over character frequency, stroke count, and orthographic structure, revealing a specific pattern of syntactic preview processing in Chinese. The results showed that, compared to the identical preview condition, the target words in the syntactically inconsistent condition exhibited significantly lower skipping rates. This finding provides direct evidence for the existence of syntactic preview effects in Chinese reading. However, it is noteworthy that none of the eye movement measures (including first-fixation duration, single-fixation duration, and gaze duration) showed significant differences between the syntactically consistent and inconsistent conditions. Moreover, syntactic information did not produce a spillover effect on the second character of the target word. This dissociated pattern—“sensitivity in skipping rates but insensitivity in fixation durations”—profoundly reflects the uniqueness of syntactic preview processing in Chinese, offering critical evidence for understanding the similarities and differences in parafoveal information processing across different language systems. Additionally, I have explicitly addressed this methodological limitation in the concluding remarks of the first paragraph in the Discussion section. It should be noted that the syntactically inconsistent condition in this study (e.g., “弹答”) simultaneously violates both grammatical and semantic rules, which may result in the observed effects reflecting a combination of syntactic and semantic anomalies. Due to the absence of inflectional and derivational morphology in Chinese, we were unable to construct experimental materials that maintained semantic plausibility while manipulating syntactic violations. Therefore, readers should interpret the findings with appropriate caution regarding this limitation.

A comparison with findings from alphabetic writing systems highlights how language-specific characteristics shape syntactic preview processing. In morphologically rich languages like English, researchers have consistently observed robust syntactic preview benefits, which manifest not only in skipping rates ([2]) but also significantly affect gaze duration and regression path duration ([20]). This advantage may stem from the rich morphological marking system in alphabetic languages, which provides reliable syntactic cues for parafoveal processing. In contrast, Chinese lacks explicit inflectional or derivational morphology, relying primarily on word order and contextual cues to convey syntactic information. This linguistic characteristic results in syntactic preview effects that exhibit “limited” and “selective” features in Chinese. Notably, the two-character verb–object phrases used in the current study demonstrated a weak but observable syntactic preview effect compared to previous studies employing single-character words ([8], [9]). This finding strongly supports the theoretical hypothesis that “lexical unit length modulates the extractability of syntactic information.” The higher semantic transparency and more explicit syntactic structure of two-character words, combined with the high part-of-speech predictability (96%) ensured by rigorous material control in this study, collectively create the necessary conditions for partial extraction of syntactic information in the parafoveal region.

From the perspective of cognitive processing mechanisms, the current findings pose significant challenges and implications for existing eye movement control models of reading. The parallelism assumption of the parallel processing model (e.g., SWIFT; [4]) is not fully supported by the observed syntactic preview effects in Chinese, as parafoveal extraction of syntactic information exhibits clear limitations—a tension with the model’s core assumptions. In contrast, the serial processing nature emphasized by the E-Z Reader model ([16]) aligns better with the “selective skipping rate effect” found in this study, where syntactic information primarily influences early saccade planning but has limited impact on subsequent lexical identification and integration processes. This processing pattern may reflect the unique attentional resource allocation in Chinese reading: in the absence of explicit morphological markers, readers must allocate more cognitive resources to foveal syntactic integration, thereby constraining the depth of syntactic processing in the parafovea. This discovery suggests that existing Chinese reading models (e.g., [7]) need to incorporate a “hierarchical syntactic processing” mechanism, where different linguistic information in the parafovea is prioritized during extraction. Specifically, orthographic and semantic information may take precedence over syntactic cues, reflecting an adaptive strategy for efficient reading in morphologically impoverished writing systems.

The theoretical significance of this study also lies in its contribution to the debate on “universality versus specificity in language processing”. There has been long-standing controversy regarding whether parafoveal processing reflects universal cognitive mechanisms or is constrained by language-specific characteristics ([11]; [27]). The syntactic preview pattern revealed in this study exhibits both commonalities with alphabetic languages (e.g., syntactic effects observable in skipping rates) and notable differences (e.g., absence of fixation duration effects). This pattern of “limited universality” supports the “language-specific modulation hypothesis”, which posits that while basic cognitive processing mechanisms are universal across languages, their specific implementations are profoundly shaped by linguistic structures. Particularly noteworthy is the observed asymmetry in Chinese preview effects: robust semantic preview benefits ([24]) but weaker syntactic preview effects. This contrast further highlights how language properties selectively modulate different aspects of information processing. Such findings underscore the need for theoretical frameworks that account for both universal cognitive constraints and language-specific adaptations in reading.

Future research should deepen and expand investigations across multiple dimensions. In terms of experimental materials, systematic examination of preview effects across different syntactic structures (e.g., subject-predicate, modifier-head constructions) is needed to establish a more comprehensive theory of syntactic preview in Chinese. Methodologically, combining high temporal-resolution neurophysiological techniques (e.g., EEG focusing on N400 components) with high spatial-resolution eye-tracking could precisely reveal the spatiotemporal dynamics of syntactic preview processing. At the theoretical level, developing a unified model capable of explaining both the robust semantic preview advantage and the constrained syntactic preview effects in Chinese is essential. This may involve postulating a “semantic priority” processing mechanism. Furthermore, systematic cross-linguistic comparisons—particularly with other non-alphabetic scripts like Japanese and Korean—would help disentangle the relative contributions of writing systems versus linguistic structures to syntactic preview. Computational modeling approaches could provide quantitative support for theoretical frameworks by parameterizing the extraction efficiency of different linguistic information. Such multi-method, cross-linguistic investigations would significantly advance our understanding of the universality and specificity of parafoveal processing.

It is important to acknowledge some limitations of this study. First, we did not employ neurophysiological techniques such as ERP or neuroimaging, which could provide additional insights into the temporal dynamics of syntactic preview processing. Second, our sample was relatively homogeneous, consisting primarily of undergraduate students, which may limit the generalizability of our findings. Future research could address these limitations by incorporating more diverse participant groups and using multimodal methods to further explore the neural basis of syntactic preview effects.

In summary, this study reveals a distinct pattern of syntactic preview processing during Chinese reading, providing important insights into the interaction between language-specific characteristics and cognitive mechanisms. These findings not only deepen our understanding of Chinese reading processes but also offer critical evidence for cross-linguistic reading theory construction. From an applied perspective, our results suggest that Chinese language education could benefit from instructional strategies that emphasize the importance of syntactic structure and word order, given the limited syntactic preview effects observed. Additionally, reading interface design could incorporate features that enhance the visibility of syntactic cues, such as using color or font size to highlight key grammatical elements, thereby potentially improving reading efficiency in Chinese. Future research should adopt a broader theoretical perspective and employ systematic, multi-method investigations to refine our understanding of parafoveal syntactic processing mechanisms, ultimately advancing more explanatory models of Chinese reading.

## Figures and Tables

**Figure 1 behavsci-15-00935-f001:**
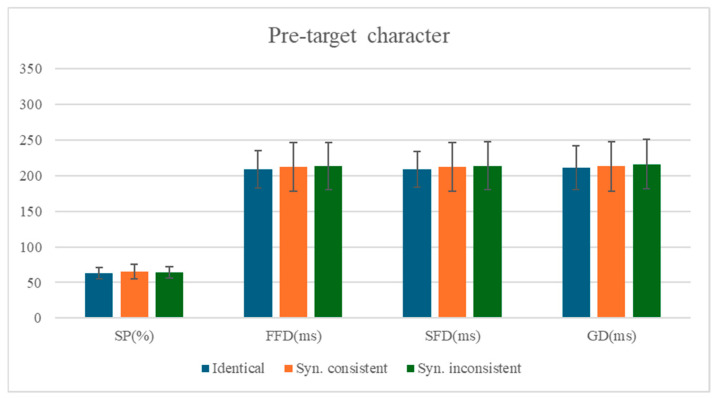
Comparison of Mean Differences Across Measures for the Pre-target character.

**Figure 2 behavsci-15-00935-f002:**
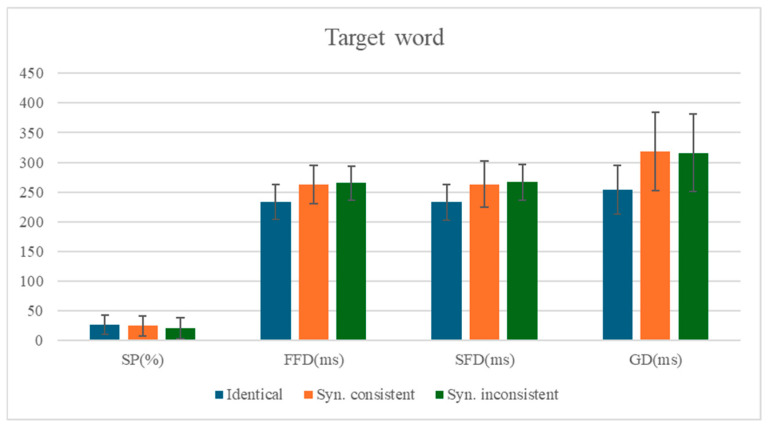
Comparison of Mean Differences Across Measures for the Target Word.

**Figure 3 behavsci-15-00935-f003:**
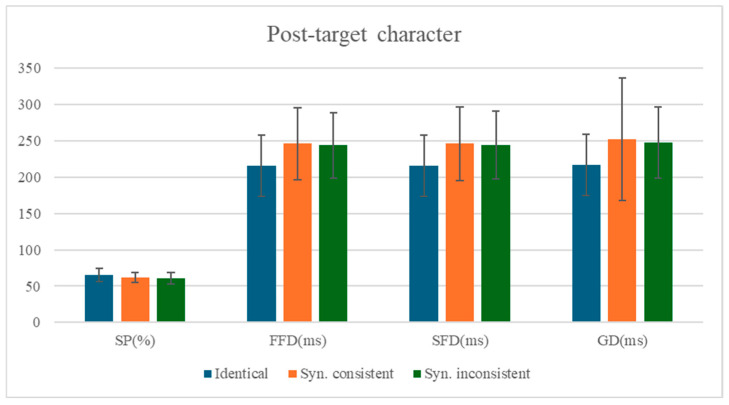
Comparison of Mean Differences Across Measures for the Post-target character.

**Table 1 behavsci-15-00935-t001:** Cross-Linguistic Comparison of Parafoveal Preview Effects.

Language Type	Semantic Preview Benefit	Syntactic Preview Benefit	Low-Level Information Preview Benefit
Alphabetic (e.g., English)	No significant	Robust	Stable
Chinese	Robust	Weak or limited	Stable

**Table 2 behavsci-15-00935-t002:** Example Stimuli (**Bold font** indicates target words and preview words).

Condition	Sentence
1 Identity	李萍平时很喜欢￤**弹琴**给小朋友们听。 Liping usually enjoys￤**playing the piano** for the children.
2 Syn. consistent	李萍平时很喜欢￤**弹窝**给小朋友们听。 Liping usually enjoys￤**playing the nest** for the children.
3 Syn. inconsistent	李萍平时很喜欢￤**弹答**给小朋友们听。 Liping usually enjoys￤**playing the answer** for the children.

Note. Condition 1 preserves both syntax and semantics; Condition 2 violates semantics only; Condition 3 violates both.

**Table 3 behavsci-15-00935-t003:** Means (M) and Standard Deviations (SD) of Eye-Movement Measures for the Pre-Target Character.

Measures	1 Identical	2 Syn. Consistent	3 Syn. Inconsistent
SP (%)	0.63 (0.08)	0.65 (0.1)	0.64 (0.08)
FFD (ms)	209 (26)	212 (34)	213 (33)
SFD (ms)	209 (25)	212 (34)	214 (34)
GD (ms)	211 (31)	213 (35)	216 (35)

Note. Standard deviations are presented in parentheses. All duration measures are reported in milliseconds(ms).

**Table 4 behavsci-15-00935-t004:** Results for the mixed-linear model analysis of pre-target character measures.

Measures	Fixed Effect	*b*	*SE*	*t*/*z*	*p*
SP	Identity	0.06	0.08	0.75	0.45
	Syntactic Preview Effect	−0.05	0.08	−0.57	0.57
FFD	Identity	0.01	0.02	0.49	0.63
	Syntactic Preview Effect	−0.01	0.02	−0.61	0.54
SFD	Identity	0.01	0.02	0.62	0.54
	Syntactic Preview Effect	−0.01	0.02	−0.36	0.72
GD	Identity	0.01	0.02	0.39	0.69
	Syntactic Preview Effect	−0.00	0.02	−0.16	0.88

**Table 5 behavsci-15-00935-t005:** Means (M) and Standard Deviations (SD) of Eye-Movement Measures for the Target Word.

Measures	1 Identical	2 Syn. Consistent	3 Syn. Inconsistent
SP (%)	0.27 (0.16)	0.25 (0.17)	0.21 (0.17)
FFD (ms)	233 (29)	263 (32)	265 (29)
SFD (ms)	233 (30)	263 (39)	267 (30)
GD (ms)	254 (41)	318 (66)	316 (65)

**Table 6 behavsci-15-00935-t006:** Results for the mixed-linear model analysis of target word measures.

Measures	Fixed Effect	*b*	*SE*	*t*/*z*	*p*
SP	Identity	−0.09	0.09	−0.94	0.35
	Syntactic Preview Effect	−0.23	0.1	−2.35	0.02
FFD	Identity	0.1	0.01	7.46	<0.001
	Syntactic Preview Effect	0.01	0.01	0.72	0.47
SFD	Identity	0.01	0.02	6.69	<0.001
	Syntactic Preview Effect	0.02	0.02	1.07	0.29
GD	Identity	0.18	0.02	10.57	<0.001
	Syntactic Preview Effect	0.00	0.02	0.27	0.79

**Table 7 behavsci-15-00935-t007:** Means (M) and Standard Deviations (SD) of Eye-Movement Measures for the Post-Target character.

Measures	1 Identical	2 Syn. Consistent	3 Syn. Inconsistent
SP (%)	0.65 (0.09)	0.62 (0.07)	0.61 (0.08)
FFD (ms)	216 (42)	246 (50)	244 (45)
SFD (ms)	216 (42)	246 (51)	244 (47)
GD (ms)	217 (42)	252 (84)	248 (49)

**Table 8 behavsci-15-00935-t008:** Results for the mixed-linear model analysis of post-target word measures.

Measures	Fixed Effect	*b*	*SE*	*t*/*z*	*p*
SP	Identity	−0.18	0.08	−2.22	0.03
	Syntactic Preview Effect	−0.05	0.08	−0.63	0.53
FFD	Identity	0.11	0.02	4.89	<0.001
	Syntactic Preview Effect	−0.02	0.02	−0.91	0.36
SFD	Identity	0.11	0.02	4.97	<0.001
	Syntactic Preview Effect	−0.02	0.02	−0.97	0.33
GD	Identity	0.12	0.02	5.35	<0.001
	Syntactic Preview Effect	−0.02	0.02	−0.95	0.34

## Data Availability

The raw data supporting the conclusions of this article will be made available by the authors on request.

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
