# Peer review of "Syntactic Information Extraction in the Parafovea: Evidence from Two-Character Phrases in Chinese"

_behavsci, 2025, doi:10.3390/bs15070935_

Round 1

Reviewer 1 Report

Comments and Suggestions for Authors
  1. Abstract

Suggestions:

  • Include the sample size in the abstract to improve methodological transparency.
  • Add brief mention of “boundary paradigm” as the core method.

  1. Introduction

Suggestions:

  • The section is dense with citations and could benefit from a figure or conceptual model to visualize the preview effects across languages.
  • Slight redundancy in re-explaining Rayner’s boundary paradigm (Lines 48–56), streamline for clarity.
  • A clearer research question or hypothesis at the end of the introduction would enhance framing.
  1. Methodology

3.1 Participants

  • Well-described, including gender distribution, compensation, and screening criteria.

3.2 Design & Materials

Suggestions:

  • The logic of syntactically “consistent” previews could be further clarified, since they are semantically implausible, could this affect processing beyond syntax?

3.3 Apparatus & Procedure

  • Technically sound, following standard eye-tracking norms.
  • Suggest including a calibration accuracy report or average deviation.
  1. Results

 Suggestions:

  • Include confidence intervals alongside all test statistics for transparency (done inconsistently).
  • For clarity, present effect sizes (e.g., Cohen's d or standardized beta) where applicable.
  • The naming of subsections (3.1, 3.2, etc.) could be more informative (e.g., “3.1 Pre-target Fixations”).
  1. Discussion

Suggestions:

  • A visual summary (e.g., table or diagram) comparing results across preview types and ROIs could greatly enhance reader comprehension.
  • Suggest briefly addressing limitations (e.g., no ERP or neuroimaging data, sample homogeneity, lack of naturalistic reading contexts).
  • A minor point: The term “semantic priority” is introduced late (Line 373), consider defining earlier if it’s central to the theoretical proposal.
  1. Conclusion and future work

Suggestions:

  • Provide a slightly more applied perspective, how might this inform Chinese language education or reading interface design?
  1. References

Suggestions:

  • Ensure consistency in formatting (e.g., missing article titles in a few entries; correct inconsistent punctuation).
  • Check DOI formatting on sources such as [3] and [26].

Reviewer 2 Report

Comments and Suggestions for Authors

Thank you for this very interesting study. I enjoyed reading the manuscript and found it a solid empirical study that makes a meaningful contribution to understanding cross-linguistic differences in reading processes. The literature reviewed was adequate, methodology was mostly rigorous, and the presentation of results and implications was well done.

However, I do have several concerns that I believe should be addressed to strengthen your discussion and conclusions:

  1. I noticed your experimental design contains what I see as a major confounding factor that isn't acknowledged in your discussion. It seems syntactically inconsistent preview condition (e.g., "弹答") would create both syntactic and semantic anomalies simultaneously. Can we determine whether the effects you observed are truly syntactic or partially driven by semantic processing? I understand this might be because it's hard to find verb-noun phrases that only violate syntax but not semantics (I can think of sth like "听唱"), and they have to be controlled for strokes, etc. However, since this is a critical issue in the experimental design, at least you should caution readers to interpret the results with care.

  2. For the same methodological issue, I find it strange that the syntactic inconsistency caused limited impact to processing, and therefore have doubts regarding the conclusion that Chinese readers have "limited sensitivity to syntactic violations in parafoveal vision". Can the authors at least down-tone the claims of relevant interpretation and discussion, and expound on why despite the seeming semantic inconsistency in the syntactic inconsistency condition, the processing effect was less than that in the pure semantic inconsistency condition. The current discussion is somewhat premature or conflicting with results of semantic inconsistency. Crucially, like previously suggested, there needs to be explicit mentioning of this methodological limitation.

  3. I am a little surprised by your choice of eye-tracking measures, which focus primarily on early-stage processing measures. Given that theoretical models like E-Z Reader (which you referenced in the discussion) suggest syntactic and semantic processing occur at later stages, I wonder why only gaze duration was selected as a late measure? I can see you cited studies that adopted regression path duration, for example; why did you not choose measures such as this for measurement validity and across-study comparison reasons?
  4. You mention SWIFT and E-Z Reader models briefly in the discussion (p. 9). I would appreciate seeing a discussion of these models/theoretical frameworks in the lit review section. 

Despite these concerns, I found your study makes a valuable contribution to our understanding of cross-linguistic differences in reading processes. I hope to see an improved version of the manuscript for a re-assessment.

Reviewer 3 Report

Comments and Suggestions for Authors

Lines 294-297 should not be included in the text. They are like 'guidelines' for the authors.

Round 2

Reviewer 2 Report

Comments and Suggestions for Authors

My comments are properly addressed. Thanks to the authors for the revision effort.

Just a note, the newly added text on p. 10 seems to be of a different font type. Please make sure the font size and type are consistent with the rest of the article.